# Group I Paks are essential for epithelial-mesenchymal transition in an *Apc*-driven model of colorectal cancer

H.Y. Chow[1,2], B. Dong[1], C.A. Valencia[3], C.T. Zeng[1], J.N. Koch[2], T.Y. Prudnikova[2] & J. Chernoff [2]

p21-activated kinases (Paks) play an important role in oncogenic signaling pathways and have been considered as potential therapeutic targets in various cancers. Most studies of Pak function employ gene knock-out or knock-down methods, but these approaches result in loss of both enzymatic and scaffolding properties of these proteins, and thus may not reflect the effects of small molecule inhibitors. Here we use a transgenic mouse model in which a specific peptide inhibitor of Group I Paks is conditionally expressed in response to Cre recombinase. Using this model, we show that inhibition of endogenous Paks impedes the transition of adenoma to carcinoma in an *Apc*-driven mouse model of colorectal cancer. These effects are mediated by inhibition of Wnt signaling through reduced β-catenin activity as well as suppression of an epithelial-mesenchymal transition program mediated by miR-200 and Snai1. These results highlight the potential therapeutic role of Pak1 inhibitors in colorectal cancer.

[1] Cancer Center, National Clinical Research Center for Geriatrics, State Key Laboratory of Biotherapy, West China Hospital, Sichuan University and Collaborative Innovation Center for Biotherapy, 610041 Chengdu, Sichuan, China. [2] Cancer Biology Program, Fox Chase Cancer Center, Philadelphia, PA 19111, USA. [3] Division of Human Genetics, Cincinnati Children's Hospital Medical Center, Cincinnati, OH 45229, USA. These authors contributed equally: H.Y. Chow, B. Dong. Correspondence and requests for materials should be addressed to J.C. (email: Jonathan.Chernoff@fccc.edu)

Colorectal cancer is the third most common cancer diagnosed in both men and women in the United States[1]. Although the incidence rate of CRC has been declining over the past decade, the mortality for advanced stages of this disease remains unchanged. Somatic inactivating mutations in the adenomatous polyposis coli (APC) gene are believed to be the crucial initiating event triggering adenoma formation, which, over time, can progress to fully malignant carcinomas. Approximately 90% of CRCs exhibit alterations of APC or other Wnt signaling pathway elements, and elevation of β-catenin levels alone is insufficient to drive the development of CRC[2,3]. In addition to Wnt pathway activation, activation of KRAS or inactivation of p53 commonly occurs as secondary genetic events, promoting tumor progression, invasion, and metastasis.

p21-activated kinases (Paks) are serine/threonine protein kinases that act as effectors for small GTPases such as Cdc42 and Rac. A growing body of evidence indicates that Group I Paks, in particular Pak1, are required for the activation of several key oncogenic signaling pathways, such as Mek/Erk, PI3K/Akt, and Wnt/βCatenin[4–6] and play central roles in promoting cell proliferation, survival, and migration[7]. Pak1 expression increases in invasive and metastatic CRC lesions[8] and correlates with a lower survival rate[9]. In addition, nuclear localization of Pak1 is associated with advanced clinical stage[10]. Knockdown of Pak1 using RNAi significantly inhibits cellular growth of CRC cells both in vitro and in vivo and enhances the chemosensitivity of CRC cells[10]. These changes are accompanied by loss of phosphorylation of β-catenin at S663 and S675, known target sites for Pak that augment β-catenin stability and transcriptional activity[11–13]. These data suggest that Group I Paks may contribute to neoplastic progression in CRC and serve as plausible drug targets in this disease.

Given the current lack of clinical Pak small molecule inhibitors, most prior studies of Pak function in cancer have relied on siRNA-mediated or shRNA-mediated gene knockdown or on Pak1 gene knockout in cells and/or animals. Such studies have shown that the Pak1 protein is a required element in transformation driven by oncogenes such as ERBB2 and KRAS[6]. However, it is well-recognized that loss-of-function induced by siRNA or gene knock out is not an optimal method to evaluate drug targets, as small molecule inhibitors usually exert their effects by inhibiting enzyme activity rather than eliminating their target. This is a particularly important consideration in the case of the Group I Paks, which have biologically important scaffold activities, as, for example, in activating the Akt pathway[14] and in focal-adhesion disassembly[15]. While this issue might in principle be addressed by using anti-Pak inhibitors, most currently available small molecule Pak inhibitors suffer several drawbacks, including lack of ideal specificity, poor pharmacologic properties, and excessive toxicity[16]. For these reasons, we wished to design a new mouse model that expresses an inhibitory peptide that closely mimics the effects of an ideal small-molecule inhibitor of Group I Paks, specifically and potently inhibiting endogenous enzyme activity without affecting Pak expression levels.

Several peptide-based inhibitors have been employed to block Group I Pak function[17–19]. These have been based on one of two designs: polyproline-containing peptides that interfere with the ability of Group I Paks to bind SH3-domain proteins such as Nck or PIX, thereby preventing membrane association and GTPase activation of the kinase, or a peptide derived from the minimal autoinhibitory domain from Pak1 (minimally, corresponding to residues 83–149), termed the PID (Pak inhibitory domain), that engages the kinase domain and inhibits activity. When expressed in cells, this peptide acts in trans to inhibit all three endogenous Group I Paks, and is not known to bind or inhibit other kinases[19,20]. In this study, we generated a targeted transgenic mouse carrying a conditionally activated PID allele at the Rosa26 locus, and showed that expression of this allele effectively inhibited the activity of Group I Paks in vivo. Conditional expression of the PID allele in an Apc-driven model of CRC was associated with blockade of carcinogenesis. By analyzing colonic epithelial cells derived from these mice, we demonstrated the existence of a Group I Pak-regulated EMT program in Apc-mutant CRC cells that involves the miR-200 microRNA family, expression of the CD44 splicing factor ESRP1, and phosphorylation of the Snai transcription factor.

## Results

**Generation of ROSA^PID transgenic mouse.** We developed a construct targeted to the mouse Rosa26 locus designed to conditionally express a GST-PID cassette[21] (Fig. 1a, b). In this construct, the transcription of GST-PID is blocked by the presence of upstream "lox-stop-lox" (LSL) sequences (Fig. 1c). Upon exposure to Cre recombinase, the "stop" sequences between the loxP sites are expected to be excised, leading to expression of the GST-PID transgene together with a downstream eGFP reporter (Fig. 1d). Chimeric animals (Tg(Rosa26-PID), hereafter referred to as PID) were identified by PCR analysis of tail biopsies DNA (Fig. 1e). When transduced with an Adeno-Cre virus, MEF cells derived from these mice showed robust expression of GST-PID and repression of endogenous group I Pak activity (Fig. 1f).

To determine if expression of the PID affected embryonic development in mice, PID mice were crossed with Tg(CDX2P-NLS-Cre) (hereafter referred to as CDX2) mice, which express Cre recombinase in intestinal tissues[22], and the intestines and other internal organs were examined in bi-transgenic CDX2;PID animals. Expression of the PID did not affect cellular intestinal architecture (CDX2 mice ($n = 8$), CDX2;PID mice ($n = 6$); Supplementary Fig. 1A) and all internal organs appeared to be normal when compared with their control littermates (Supplementary Fig. 1B). As anticipated, no changes in proliferation (BrdU) or apoptosis (cleaved caspase 3) was detected by immunohistochemistry (Supplementary Fig. 1A). These results indicated that conditional expression of the PID peptide does not radically alter mouse gastrointestinal development or basal signaling.

**Effect of the PID peptide in a APC-driven CRC mouse model.** To investigate the effects of Pak inhibition on oncogenesis in vivo, progeny from CDX2;PID and APC;PID (APC^{loxP/+};PID) were intercrossed to generate mice with the genotype CDX2P-NLS-Cre; APC^{Δ/+};ROSA26-PID (denoted as CDX2;APC;PID) and control littermates (CDX2;APC^{Δ/+}, denoted as CDX2;APC), in which expression of these transgenes are restricted to epithelium primarily in the large intestine and distal regions of the small intestine. In this model exon 14 of the Apc^{loxP} allele is targeted leading to a frameshift mutation at codon 580 and a truncated APC protein[23].

Following 8 months of observation, CDX2;APC;PID mice ($n = 15$) appeared healthy and gradually gained weight. In contrast, their CDX2;APC littermates ($n = 21$) were lighter, likely as a consequence of the intestinal obstruction by tumors (Fig. 2a). No statistically significant difference in body weights was revealed in control cohorts, irrespective of the expression of the PID (Supplementary Fig. 2A).

All CDX2;APC transgenic animals lacking PID expression developed numerous adenomas in the large intestine coupled with some in the small intestine, consistent with previous reports[22]. In contrast, one third of the CDX2;APC;PID mice remained free of tumors, and the number of tumors per mouse in this cohort was less than that seen in mice lacking the PID

transgene (Fig. 2b). Interestingly, the number of adenomas was 40 and 65% less in small and large intestine in *CDX2;APC;PID* mice, respectively (Fig. 2c). Tumors dissociated from all mice appeared either a stalked (polypoid adenoma) or a sessile morphology; however, adenocarcinomas only appeared in mice lacking PID

expression (Fig. 2d). The average tumor area was significantly reduced in *CDX2;APC;PID* mice (3 mm$^2$) compared with those found in *CDX2;APC* mice (14 mm$^2$) (Fig. 2e, f).

Immunohistochemical analysis of adenomas from *CDX2;APC* mice showed that these lesions were highly proliferative, as

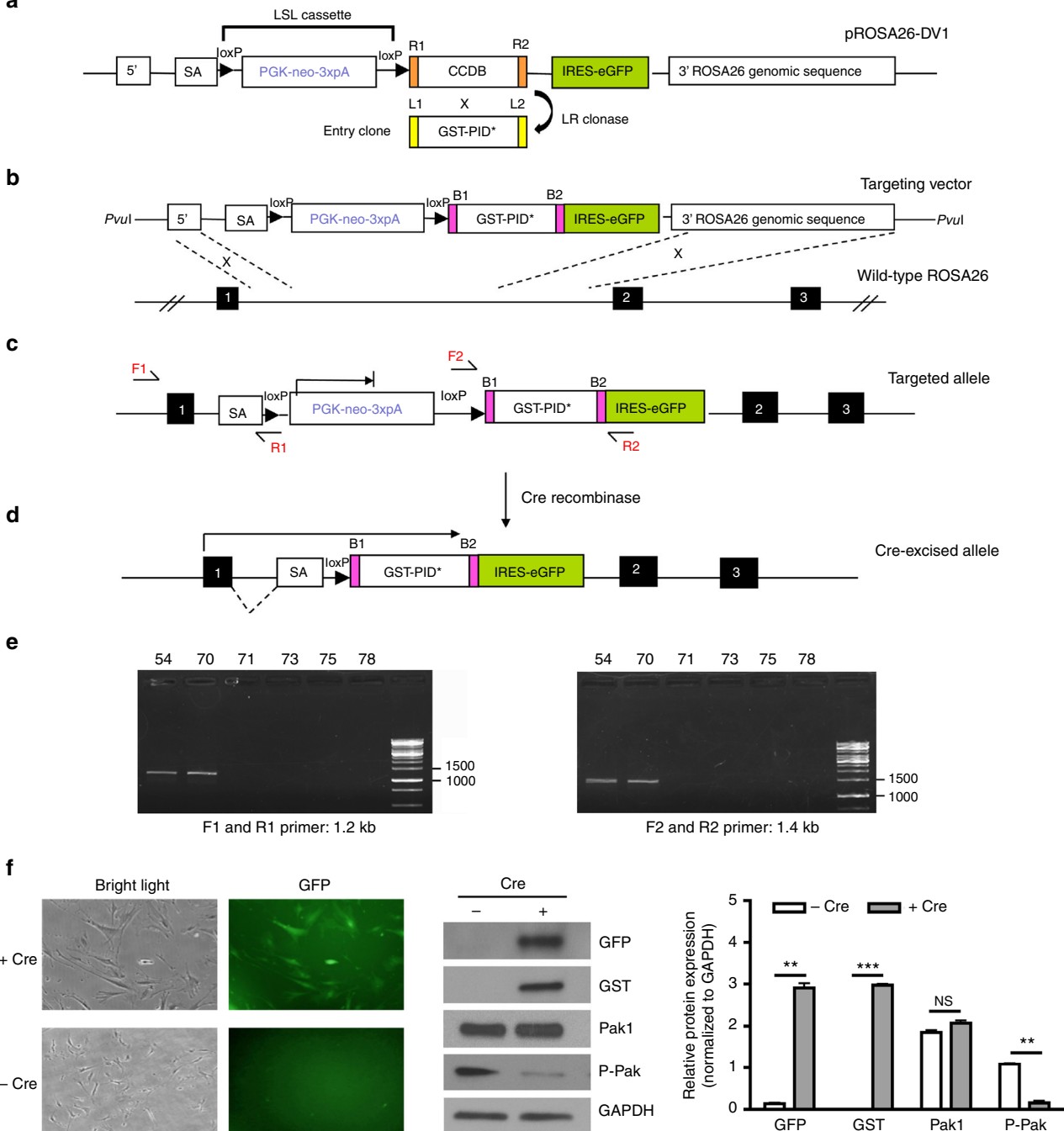

**Fig. 1** Generation and analysis of conditional *Rosa26*-promoter-based expression alleles. **a** LR reaction performed between the pROSA26-DV1 vector and pEntry clone containing *GST-PID\** fragment to generate the *Rosa26* targeting vector. SA is spice acceptor, PGK is phosphoglycerate kinase 1 promoter, and 3xpA is multimerized polyadenylation sequence in which loxP-PGK-neo-3xpA-loxP formed a LSL cassette. **b** Homologous recombination occurred between exon 1 and 2 of wild-type Rosa26 locus in G4 ES cells after electroporation. Black boxes represent the exons located at *ROSA26* locus. **c** Knock-in targeted allele analyzed by PCR using both external primers (F1 and R1) and internal primers (F2 and R2). **d** Cre-mediated excision of intervening *loxP* flanked PGK-neo-3xpA (STOP) cassette resulted in the *Rosa26*-locus-based expression of an exon1-GST-PID\*-IRES-eGFP bi-cistronic fusion transcript. **e** Representative result of genotyping PCR analysis of tail biopsy DNA detecting the presence of fusion transcript by both external primers (F1 and R1, 1.2 kb) and internal primers (F2 and R2, 1.4 kb), in which #54 and #70 were mice revealing positive results. **f** Phase contrast image (left side) and green fluorescence image (right side) showing the same field of view of MEF cells derived from PID mice after Adeno-cre virus infection. Representative western blot of GFP, GST, Pak1, Phospho-Pak expression level in MEF cells after infection of Adeno-cre virus. Relative protein expression was quantified and normalized to GAPDH

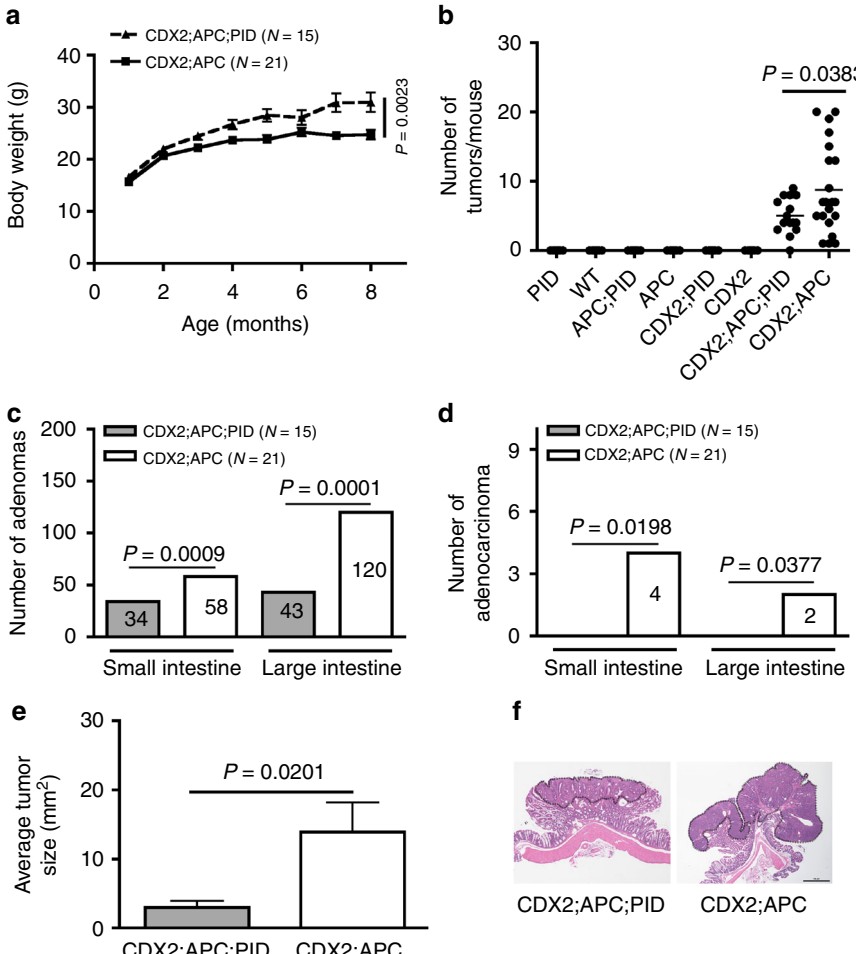

**Fig. 2** Inhibition of Group I Paks activity by PID retards tumor progression. **a** Body weights of *CDX2;APC;PID* (experimental, *n* = 15) and *CDX2;APC* (control, *n* = 21) mice was measured once every two weeks for 8 months. **b** Number of tumors per mouse were counted from control and experimental mice at 8 months of age, at which time these animals were sacrificed. **c** Adenoma number found in the intestinal tracts from both cohorts of animals at 8 months of age. **d** Adenocarcinoma number found in the intestinal tracts from both cohorts of mice at 8 months of age. **e** Tumor size was measured on paraffin-embedded sections stained for H&E collected from both groups of animals. **f** Representative H&E staining image of colon tumor dissected from *CDX2;APC; PID* and *CDX2; APC* mice (x20 magnification). Bar, 100 µm

characterized by Histone H3-positive cells, while adenomas from *CDX2;APC;PID* mice had lower levels of Histone H3-positive cells. In addition, strong activation of Pak and Akt was observed in adenomas from *CDX2;APC* mice, compared to weak staining exhibited in adenomas from *CDX2;APC;PID* mice (Supplementary Fig. 2B). Taken together, the noted signaling differences and lack of progression to adenocarcinoma in *CDX2;APC;PID* mice suggests that Group I Pak kinase activity is a key factor in tumorigenesis and the expression of PID peptide is sufficient to impede tumor progression in this CRC mouse model.

**Influence of PID expression on cell signaling pathways**. To address the molecular mechanisms by which the PID repressed proliferation and tumorigenesis in mice, we established 6 primary colonic cell lines from adenomas from both cohorts (*CDX2;APC; PID* and *CDX2;APC*, 3 cell lines for each genotype). Primary cells expressing the PID peptide (*PID+* cells) retained their cobble-stone epithelial morphology whereas *PID-* cells acquired a fibroblast-like appearance with more cell protrusions (Supplementary Fig. 3A). In agreement with our IHC data, proliferation was suppressed in *PID+* cells compared to *PID-* cells (Fig. 3a), while apoptosis did not change in these cells irrespective of *PID*

expression (Supplementary Fig. 3B). In addition, we also examined the invasive and migration abilities of these primary cell lines. These assays showed a dramatic repression in invasiveness and motility in *PID+* cells (60% and ~72% reductions, respectively) (Fig. 3b, c).

Pak1 has been implicated in modulating MAPK signaling, via phosphorylation of c-Raf and Mek1;[24] Akt signaling, via scaffolding effects on PDK1;[14] and Wnt signaling, through phosphorylation of β-catenin at C-terminal sites[12]. We found that expression of *PID* was associated with downregulation of both MAPK and Akt activity as well as a reduction in β-catenin phosphorylation. In addition, suppression of cyclin D1 expression level was also noted in *PID+* cells (Supplementary Fig. 4A), consistent with repression of MAPK and β-catenin signaling. Similar effects were observed when *PID-* cells were treated with Frax-1036, a relatively specific Group I Pak inhibitor[25–27]. Frax-1036 potently blocked Pak activation, as assessed by phospho-Pak blot, as well as Mek activation and β-catenin expression (Supplementary Fig. 4B–D). To further determine the effect of small molecule Pak inhibitors on CRC tumor growth, Frax-1036, as well as a second, structurally unrelated Pak inhibitor (G5555)[28], was tested in CRC xenografts using Wnt pathway mutant SW48 cells[29]. Mice were divided into four groups and,

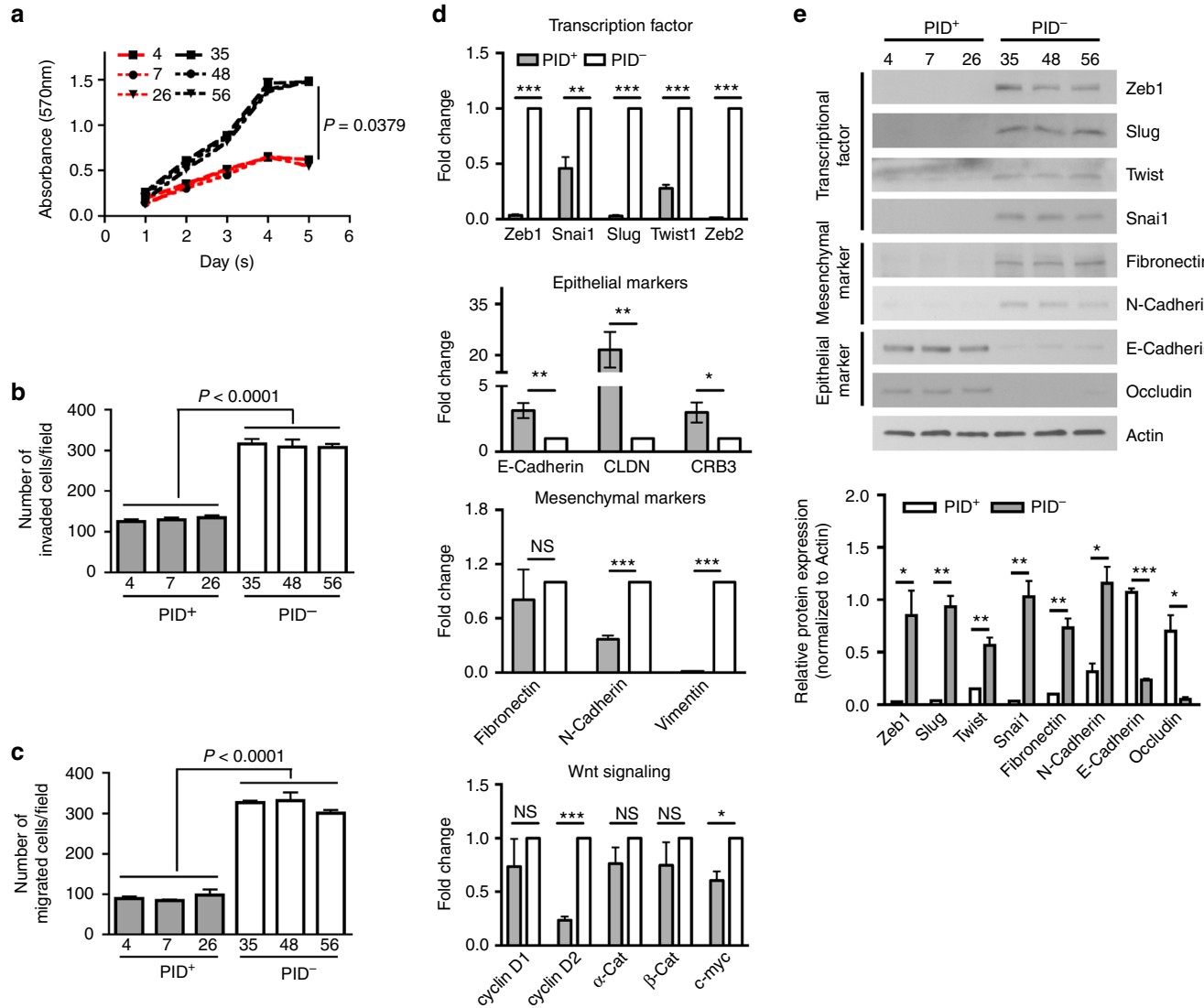

**Fig. 3** Expression of PID inhibits invasiveness and motility by suppressing EMT. **a** Cell viability of primary colon epithelium tumor cells derived from *CDX2; APC;PID* (#4, 7 and 26; red line) and *CDX2;APC* (#35, 48 and 56; black line) mice was measured by MTT assay. **b** Invasion assay was performed on *PID*+ (4, 7, and 26) and *PID*⁻ (35, 48, and 56) primary cells after 19 h incubation and the number of cells was counted using microscope in five random fields per chamber. **c** The number of migrated primary cells on both *PID*+ (4, 7, and 26) and *PID*⁻ (35, 48, and 56) was counted in five random fields per chamber under phase contrast microscope. **d** Real-time PCR analysis measured relative expression level of EMT genes in *PID*+ and *PID*⁻ primary cells, including *Zeb1, Zeb2, Snai1, Slug*, and *Twist1* (Transcription factors), *Fibronectin, N-Cadherin* and *Vimentin* (Mesenchymal markers), *E-Cadherin, CLDN* and *CRB3* (Epithelial markers) and genes related to Wnt signaling pathway (*cyclin D1, cyclin D2, α-Cat, β-Cat* and *c-myc*). **e** Representative immunoblot analysis of EMT markers on *PID*+ and *PID*⁻ primary cells and relative expression was quantified the intensity by Image J

when xenograft tumors reached 40 mm³, dosed daily with either of the Pak inhibitors or vehicle. Mice were weighed and tumor size measured every two days for two weeks. As shown in Supplementary Fig. 4E, dramatic suppression of tumor growth was observed in mice treated with Frax-1036, with average tumor volumes reduced by 70% (from 335 to 99.78 mm³). Even more impressive effects were noted in mice dosed with G-5555, with tumor volumes suppressed by ~80% (Supplementary Fig. 4F). Collectively, these findings demonstrated that the antitumor effects of transgeneic PID expression were similar to those of small molecule Group I Pak inhibitors. The reduced rate of tumor growth, but lack of tumor regression, seen in Pak-inhibited mice is consistent with these agents exerting a primarily anti-proliferative as opposed to an apoptotic effect, as suggested by the in vitro data presented in Supplementary Fig. 3.

Because the transgenic PID peptide used in these studies contains a GST tag and has been reported to have effects independent of Pak inhibition[30], we further tested the specificity of this molecule. *PID*⁻ cells were infected with a retrovirus carrying either no insert (IRES), GST alone (GST), GST-PID-E129K (GST-PID*), or GST-PID-E129K-L107F (GST-PID* LF), a mutant that lacks the ability to inhibit Pak[19]. Expression of GST-PID was associated with a reduction in cell proliferation as compared to control cells or cells expressing GST alone or GST-PID L107F (Supplementary Fig. 5A, B). These data suggest that growth inhibition is due to Pak inhibition by the PID.

**Alterations in transcriptome upon Group I Pak inhibition.** To assess the global molecular effects of Pak inhibition in *Apc*-null CRC cells, we next explored the effect of repressing Pak activity

on transcription. Total RNA was extracted from $PID^+$ and $PID^-$ cells and RNA-sequencing was preformed, with subsequent enrichment analysis (Supplementary Fig. 6A), pathway analysis of differentially expressed genes (Supplementary Fig. 6B, C and Supplementary Fig. 7), and qPCR validation for selected mRNAs. We found a prominent change in mRNA expression among genes regulating EMT, such as *Snai* (Supplementary Fig. 7A), which was downregulated in $PID^+$ cells. In addition to these genes, PID expression was associated with gain in expression of epithelial related genes (*E-Cadherin*, *CLDN*, and *CRB3*) and loss of mesenchymal markers (*N-Cadherin* and *Vimentin*) (Fig. 3d). Furthermore, the expression levels of several Wnt signaling pathway factors and their downstream targets (*Ccnd1* and *c-myc*) were decreased in $PID^+$ cells, with the exception to *Wnt10a*, whose expression was increased (Fig. 3d and Supplementary Fig. 8A). In addition to EMT genes, expression of genes involved in colonic epithelial self-renewal (*Lgr5*, *Msi1*, and *Bmi1*) was suppressed in $PID^+$ cells (Supplementary Fig. 8B), in accordance with the idea that cells undergoing EMT gain stem cell-like properties[31].

To verify the phenotypes suggested by these RNA profiles, we examined the state of key EMT proteins by immunoblot. The expression of *Zeb1*, *Snai1*, *Snai2* (*Slug*), and *Twist* was nearly undetectable in $PID^+$ cells, whereas it was readily apparent in $PID^-$ cells. Moreover, cells with active Pak (*i.e.*, $PID^-$ cells) had much higher expression of Fibronectin and N-Cadherin. In contrast, the expression level of E-Cadherin and Occludin was nearly undetectable in $PID^-$ cells (Fig. 3e).

These data were corroborated by results of IHC staining of CRC tumor samples from CDX;APC;PID- and CDX;APC;PID+ mice (Supplementary Fig. 8C). For example, we found that E-cadherin, ESRP1 staining was greatly increased in tumors from CDX2;APC;PID mice, whereas Msi1 was decreased in these tissues, relative to tumor tissue from animals lacking PID expression. Interestingly, Bmi1 expression was not apparent in adenomas from either group of animals, whereas patches of strong immunoreactivity were found in infiltrating cells from CDX2;APC mice. It is unclear whether these Bmi1 positive cells represent infiltrating inflammatory cells or derive from epithelial CRC tumor cells. Caspase 3 staining was not prominent in tumor tissue from either CDX;APC;PID− or CDX;APC;PID + mice. Collectively, these data are consistent with the idea that suppression of Group I Pak activity can reverse and/or prevent EMT and the acquisition of transit amplifying cell-like characteristics in *Apc*-null colonic epithelial cells.

**Pak activity modulates expression of the miR-200 family**. Given the prominent effects of Pak activity on the expression of a suite of genes that regulate EMT, we asked if these effects might be mediated by master switches of this process. First, we focused on the miR-200 family, which has been recognized as a potential tumor suppressor via its inhibitory effects on *Zeb1/2* expression[32]. We found that $PID^+$ cells, which have a epithelial-like phenotype, had elevated transcription of all five miR-200 members, from a >10-fold increase for miR-200a and miR-200b to an ~2-fold increase for miR-200c and miR-141 (Fig. 4a). To assess whether this increase in miR-200 expression was required for the epithelial-like properties of $PID^+$ cells, we reduced miR-200 expression by transfecting $PID^+$ cells with a locked nucleic acid (LNA)-based miR-200 inhibitor. Cells transfected with such an inhibitor were significantly more invasive and migratory, especially cells that expressed miR-141/miR-200a (Fig. 4b, c). Notably, these effects were confined to increased cell motility and invasiveness; none of the miR-200 inhibitors promoted cellular growth (Supplementary Fig. 8D). Consistent with this phenotype,

these changes were accompanied by a marked reduction in mRNA expression for epithelial genes, including complete loss of E-Cadherin expression in cells transfected with either single or combined miR-200 inhibitors (Fig. 4d, e).

As the miR-200 family is thought to be essential for programming cells to an epithelial state, we also investigated if ectopically expressing miR-200 would cause $PID^-$ cells, which have a mesenchymal-like phenotype, to undergo MET, regaining an epithelial phenotype. We therefore generated new $PID^-$ cells that constitutively expressed either the *miR-200b-200a-429* cluster or the *miR-200c-141* cluster by transducing these cells with an appropriately designed lentiviruses, confirming their expression by RT-PCR (Supplementary Fig. 8A). Enforced expression of either *miR* cluster increased the transcriptional levels of the epithelial markers *Crb3* and *E-Cadherin*, especially in cells transduced with the *miR-200b-200a-429* lentivirus (Fig. 4f). Conversely, N-Cadherin protein levels were reduced by >90% in cells transduced with either *miR* cluster (Fig. 4g). These results also suggest that inactivation of Paks upregulates miR-200 expression, leading to restriction of EMT by modulating expression of epithelial-related gene.

**Group I Paks regulate CD44 splicing**. To further probe EMT signaling pathways that are regulated by Pak activity, we analyzed the levels of CD44 variants in these primary CRC cells. The *CD44* gene has been found to splice differentially during EMT and a switch between isoforms is required for proper EMT induction[33,34]. The level of *CD44* mRNA isoforms was monitored by qPCR using different sets of primers and composition of these transcripts was accomplished by using exon-specific RT-PCR as well (Supplementary Fig. 8B, C). $PID^+$ cells expressed at least five-fold higher levels of *CD44v4-10* variants compared to controls, whereas the *CD44* standard isoform (*CD44s*), which is associated with a mesenchymal phenotype in CRC cells, was predominantly expressed in $PID^-$ cells (Fig. 5a). Exon skipping primers specific for *CD44s* clearly demonstrated an amplified product in $PID^-$ cells only (Supplementary Fig. 9C). The shift in isoform expression from CD44v to CD44s was confirmed by immunoblot with CD44 antibodies that recognize both the variant and standard isoforms. Cells with inactive Paks expressed the CD44v variant, but completely lacked the CD44s isoform (Fig. 5b). There was no statistically significance difference in total *CD44* mRNA levels, indicating that cells adopt alternative splicing regulation to generate CD44s isoform rather than changing overall transcription (Fig. 5a).

Two splicing factors, epithelial splicing regulatory protein 1 (ESRP1) and heterogeneous nuclear ribonucleoprotein M (hnRNPM), have been reported to regulate the CD44 isoform switch[33,35]. To investigate if these splicing factors contributed to CD44s expression, we fractionated four of the mouse-derived colonic epithelial cell lines, two from each genotype, into subcellular components and examined ESRP1 and hnRNPM expression and localization. ESRP1 was found in both the cytoplasmic and nuclear compartments of $PID^+$ cells; however, it was not detectable in either compartment in $PID^-$ cells (Fig. 5c). This finding is consistent with previous report, as downregulation of ESRP1 is found in cells with high CD44s levels[34]. In contrast, no difference in hnRNPM expression was observed between $PID^+$ and $PID^-$ cells, and expression was detected in the nuclear fraction only. We also examined the expression level of PCBP1, another splicing factor that has recently been suggested to regulate CD44 variant splicing[36]. As with hnRNPM, PCBP1 was localized both in both the cytoplasm and nucleus, but no alteration of expression or localization was found between $PID^+$ and $PID$ cells (Fig. 5c). These results suggest that ESRP1, but not

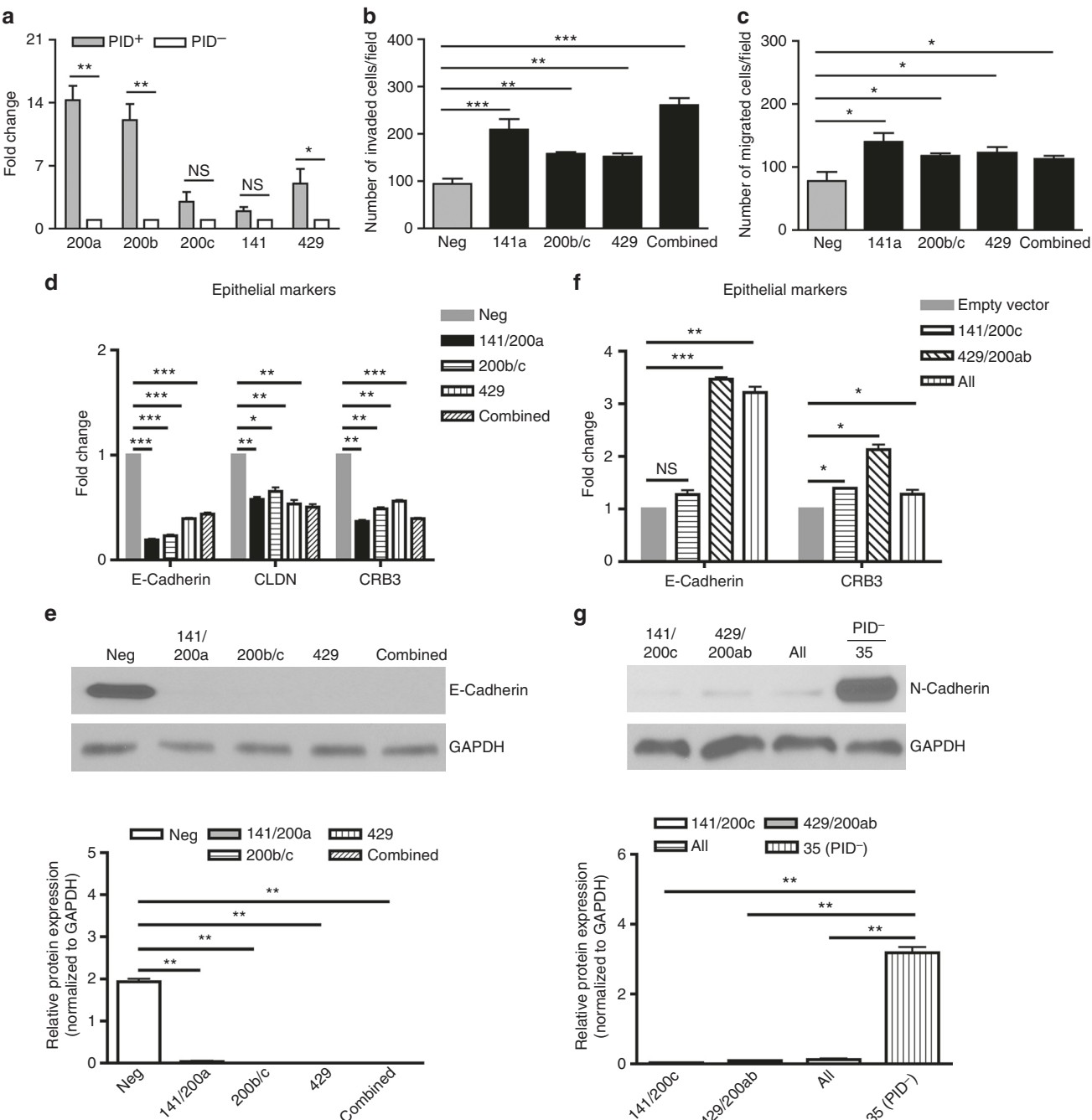

**Fig. 4** Upregulation of epithelial markers via miRNA-200 family. **a** Changes in pre-miRNA-200 relative expression levels in $PID^+$ and $PID^-$ primary cells, as measured by real-time PCR. **b** $PID^+$ cells were transfected repeatedly every 3 days with either LNA-scrambled control (Neg) or LNA-200 oligonucleotide, then subjected to invasion assay. **c** Migration assay of $PID^+$ primary cells transfected with either LNA-Scrambled control (Neg) or LNA-200 oligonucleotide. **d** qRT-PCR analysis of relative expression level of epithelial markers in $PID^+$ primary cells transfected with either LNA-Scrambled control (Neg) or LNA-200 oligonucleotide. **e** Western blotting analysis and quantification of E-Cadherin in $PID^+$ primary cells either harboring LNA-Scrambled (Neg) or LNA-200 oligonucleotide. GAPDH served as a loading control. **f** Changes in gene expression of epithelial markers in $PID^-$ primary cells transduced with lentivirus bearing either *miR-200b-200a-429* cluster or *miR-200c-141* cluster, as measured by real-time PCR. **g** Western blotting analysis and quantification of N-Cadherin in $PID^-$ primary cells (#35) expressing either *miR-200b-200a-429* cluster or *miR-200c-141* cluster. GAPDH served as a loading control

hnRNPM or PCBP1, is a major determinant of the CD44 isoform shift in these cells and that its expression is regulated by Group I Paks.

**PID expression attenuates EMT by decreasing Snai1 activation**. Expression of ESRP1 is regulated by Snai1, a transcription factor

that promotes EMT[37]. Pak1 has been reported to regulate the function of Snai1 via phosphorylation and control of its sub-cellular localization[38]. To gain further insight into the role of Pak in EMT, we analyzed the phosphorylation status of Snai1 in $PID^+$ and $PID^-$ cells. As shown in Fig. 5c, Snai1 was phos-phorylated only in the nuclear fraction extracted from cells with active Pak (i.e., $PID^-$ cells). Since Snai1 is degraded via PKD1

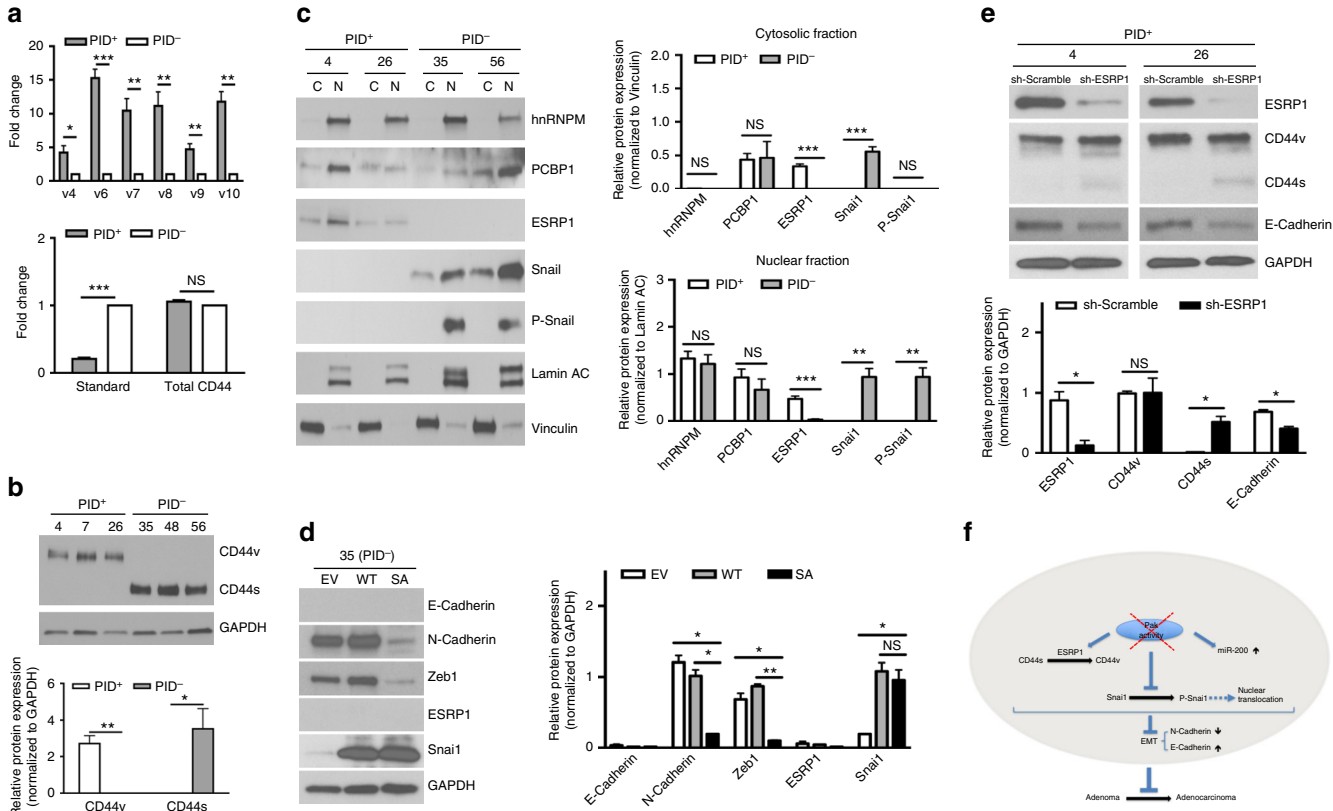

**Fig. 5** Alternative splicing of CD44 contributes to EMT in *PID*-expressed cells. **a** Real-time PCR analysis showed relative expression levels of standard isoform, total and variants of CD44 in *PID*+ and *PID*- primary cells. Results were normalized to *GAPDH*. **b** Immunoblot analysis and quantification of CD44 isoform in *PID*+ (4, 7, and 26) and *PID*- (35, 48, and 56) cells. GAPDH served as the internal control. **c** Immunoblot analysis showed protein levels and localization of hnRNPM, PCBP1, ESRP1, Snai1 and P-Snai1 in *PID*+ (4 and 26) and *PID*- (35 and 56) primary cells. Lamin A/C and Vinculin served as loading control for cytoplasmic (C) and nuclear (N) fraction, respectively. Graphs represented quantification of proteins in nuclear and cytosolic fraction. **d** Immunoblot analysis and quantification of E-Cadherin, N-Cadherin, Zeb1, ESRP1, and Snai1 in *PID*- primary cells either transduced with retrovirus bearing empty vector (EV), wild-type Snai1 (WT) or virus expressing mutant Snai1-S246A (SA). GAPDH serve as a loading control. **e** Immunoblot analysis and quantification of ESRP1, CD44, E-Cadherin and GAPDH in *PID*+ primary cells either transduced with sh-Scramble or shRNA against ESRP1. **f** Schematic illustration of invasiness inhibition by PID peptide

phosphorylation and ubiquitination mediated by p53[39,40], we next questioned whether proteasome degradation was responsible for downregulating Snai1 protein in *PID*+ cells. However, treatment of cells with the proteosomal inhibitor MG132 did not restore the expression of Snai1 (Supplementary Fig. 8D). These results suggest that Pak inhibits Snai1 expression primarily at the transcriptional rather than the post-translational level.

Phosphorylation of Snai1 at serine residue 246 has been identified as a key target site of Pak1 that activates transcriptional function[38]. To ask if the EMT characteristics described above were mediated by Snai1 phosphorylation, we infected *PID*- cells with retrovirus bearing either empty vector (EV), wild-type Snai1 (WT) or a Snai1 mutant that cannot be phosphorylated by Pak1 (Snai1[S246A], hereafter termed Snai1 SA). Expression of this mutant in *PID*- cells resulted in a strong repression of N-Cadherin and Zeb1 (Fig. 5d). In addition, we found that the nuclear accumulation of Snai1 was decreased in cells with Snai1 SA mutant compared with cells infected with WT Snai1 (Supplementary Fig. 8E). Strikingly, we did not detect re-expression of E-Cadherin and ESRP1 in Snai1 SA-transfected cells, implying that Snai1 probably not involve in modulation of their expression (Fig. 5d).

As ESRP1 is one of the mediators involving in an epithelial phenotype maintenance, we asked whether depletion of ESRP1 could promote the expression of EMT markers in *PID*+ cells.

We analysed the status of two epithelial markers in *PID*+ cells transduced with lentiviruses bearing shRNA against ESRP1. Upon depletion of ESRP1, there was a slight but readily apparent increase in expression of the CD44s isoform (Fig. 5e). ESRP1 depletion was also accompanied by a decrease in E-Cadherin expression. Taken together, these results indicate that Group I Pak regulates the expression of ESRP1, which may contribute to its effects on EMT.

## Discussion
In this study we demonstrate the utility of *PID* transgenic animals, which conditionally express a specific peptide inhibitor of Group I Paks, to study the role of these kinases in a genetically engineered cancer mouse model of colon cancer. When activated by Cre recombinase, we found that the *PID* transgene effectively suppressed endogenous Group I Pak activity in vivo, and that such reduction resulted in a marked reduction in the number and progression of adenomas to carcinoma in an *Apc*-driven model of CRC. Tissues from *PID*-expressing colonic epithelia showed a strong suppression of Erk and Akt signaling activity. In addition, cells from such animals showed reduced expression of transcriptional factors associated with EMT, increased expression level of the miR-200 family, increased expression of ESRP1, a switch in CD44 isoforms, and loss of phosphorylation of the

master EMT regulator Snai1. Together, these results highlight that Group I Pak kinase activity is a key factor in regulating the epithelial signaling program during carcinogenesis and also support the idea that small molecule inhibitors of Group I Paks might be useful in the treatment of CRC.

Pak1 in particular is suspected of playing a driver role in several different cancers, and animals bearing deletions in *Pak1* have been used to study its roles in malignancy. *Pak1*[-/-] mice are viable but exhibit immune deficiencies[41] and, in genetically engineered mouse cancer models, deletion of *Pak1*[-/-] has been shown to slow tumorigenesis induced by oncogenic forms of *ERBB2* or *KRAS*[13,42]. However, it is uncertain whether these beneficial effects are mediated by loss of kinase function or loss of scaffolding functions, or a combination of the two. The *PID* transgenic mouse, in which endogenous Group I Paks are inhibited but not absent, potentially provides a more realistic system to delineate their biological functions and to predict outcomes of treatment with specific Pak-specific small molecule inhibitors. The PID derived from Pak1 is known to bind only two classes of protein: Group I Paks and the fragile × mental retardation proteins FXR1 and FMR1[20]. Binding to FXR1/FMR1 is associated with alterations in cell cycle, and may explain off-target effects of Pak1-derived PID expression that have been observed in some settings[30]. However, binding to FXR1/FMR1 only occurs with the Pak1-derived PID, not the closely related PIDs from Pak2 or Pak3. This specificity is related to the presence of an acidic residue (glutamic acid) in the Pak1 PID that is represented by a basic residue (lysine) at the equivalent position in Pak2 and Pak3[20]. Our transgene encodes a Pak1-derived PID bearing a synthetic E129K mutation, and thus does not bind FXR1/FMR1. This peptide represents the most specific Pak inhibitor known, with no recognized off-targets among protein kinases.

Several small molecule Pak inhibitors have been described over the past few years. One of these drugs, the pan-Pak inhibitor PF-3758309, entered clinical trials; however, the development of this compound was discontinued because of unsatisfactory pharmacokinetics and adverse side effects[27]. Frax-1036, a more specific inhibitor of Group I Paks only, has been shown to inhibit cellular proliferation and to induce of apoptosis in several cancer cell types[28,43,44]. Recently, Ndubaku et al. modified Frax1036 to create a more specific inhibitor, G-5555, by relocating an amine group to improve its stability and potency[45]. These developments, plus the recent description of an isoform-specific small molecule Pak inhibitor that can discriminate between Pak1 and Pak2, suggest that such agents may find clinical utility in treating various human malignancies that require Pak function, including CRC.

Several mechanisms have been proposed to account for the role of Group I Paks in tumorigenesis. These include (i) phosphorylation of c-Raf and Mek1, augmenting Erk activation, (ii) C-terminal phosphorylation of β-catenin, stabilizing this transcriptional regulator and promoting its entry into the nucleus, (iii) a scaffold-mediated effect on Akt activation, and (iv) effects on proteins that regulate cytoskeletal dynamics and structure[6]. Given the important of Wnt signaling in CRC, the effect of Pak on β-catenin has been studied in some detail in this disease. In cell-based colon cancer models, loss of Pak by siRNA is associated with decreased proliferation and invasion, and decreased β-catenin activity. This effect is thought to be mediated by C-terminal, stabilizing phosphorylation of β-catenin by both Group I and Group II Paks[12,46–52]. Consistent with these publications, we found that tissues and cells derived from PID-expressing *Apc*-null mice showed reduced activity of Erk, Akt, and β-catenin (Supplementary Fig. 4A). In addition, we found that PID expressing cells had a markedly more epithelial morphology than control cells (Supplementary Fig. 3A). While some aspects of this phenotype might be explained by the effects of Pak on β-catenin activity, our data suggest a more direct regulation of EMT via miR-200 and Snai. *PID*[+] cells had elevated levels of miR-200, and this increased expression might account for the upregulation of ESRP1 and subsequent switch of CD44s to the CD44v isoform[53,54] (Fig. 5f). In addition, we confirmed that Pak1, directly or indirectly, regulates Snai1 through phosphorylation (Fig. 5c, d), and showed that preventing this event by expressing a non-phosphorylable form of Snai1 reverted *PID*[+] cells to a mesenchymal state (Fig. 5d).

Snai1 has been reported to interact with β-catenin in the nucleus to promote transcription of Wnt targets[55]. It is thus plausible that the beneficial effects of suppressing Group I Paks in the setting of *Apc* deletion are mediated by reductions in both EMT (via miR-200 and Snai1) and Wnt (via β-catenin) signaling pathways. Because Group I Paks play regulatory roles in multiple pathways that contribute to the transformed phenotype in *Apc*-mutant CRC, these kinases may represent particularly attractive therapeutic targets in this disease.

## Methods

**Vector construction.** The Gateway-compatible pROSA26-DV1 vector was obtained from Dr. Jody Haigh[21]. A GST-PID fragment (bearing a *Pak1*[E129K] mutation to abrogate FMR1 binding, termed *PID**) was cloned into a pEntry vector after PCR and gel purification, using the following oligonucleotide pair: GCGGCCGCCACCATGGCCCCTATA (Forward), GGCGCGCCTCATGACT TATCTGTA (Reverse), to amplify the human PID gene fragment from pGEX-CRIB[56]. The LR reaction was performed using Clonase[TM] Enzyme Mix (Life Technology) according to the manufacture's instruction. A positive clone (pROSA26-GST-PID*-IRES-eGFP) was linearized by *Pvu*I and electroporated into G4 ES cells.

**ES cell culture and aggregations.** The G4 ES cell line[57] was grown and manipulated at 37 °C in 5% $CO_2$ on mitomycin C-treated mouse embryonic fibroblasts in high-glucose DMEM supplemented with 15% ES-graded fetal bovine serum (FBS), 2 mM L-glutamine, 1 mM sodium pyruvate, 0.1 mM non-essential amino acids, 0.1 mM 2-mercaptophenol and 2000 U/ml recombinant leukemia inhibitory factor (LIF).

**Generation of transgenic mice.** Twenty-four hours after electroporation, G418 (200 μg/ml) was added to medium for 10 days to select for neomycin-resistant cells. 200 of ES cell clones were picked and genomic DNA was obtained for screening positive clone using F1 (TAGGTAGGGGATCGGGACTCT) and R1 (GCGAA GAGTTTGTCCTCAACC), F2 (CCCATCAAGCTGATCCGGAACC) and R2 (GTGAACAGCTCCTCGCCCTTG) primer pairs to generate a 1.2 kb and 1.4 kb PCR product, respectively.

*Rosa26* targeted ES cells were utilized in diploid embryo for ES cell aggregation experiment. Female mice were super-ovulated and diploid embryos were obtained by flushing from the oviduct. ES cells were gently trypsinized in 0.25% trypsin/ EDTA (Life Technology) to break up into small clumps with 8–15 cells and placed next to the embryos for cell aggregation. Blastocyst stage embryos were transferred into pseudo-pregnant female mice.

All animal experiments were approved by the Fox Chase Cancer Center Institutional Animal Care and Use Committee (IACUC) and carried out according to NIH-approved protocols in compliance with the guide for the Care and Use of Laboratory Animals. Genomic DNA of pups was prepared from tails for detecting existence of transgene by both external primer (F1 and R1) and internal primer (F2 and R2). The adult agouti bearing ROSA26 targeted allele was crossed with *CDX2P-NLS-Cre* and *APC*[loxP/+] transgenic mice (both are C57BL6/J) separately to generate *CDX2P-NLS-Cre;ROSA26-PID* and *APC*[loxP/+];ROSA26-PID colonies, termed *CDX2;PID* and *APC;PID* mice, respectively. Progeny from these colonies were subsequently bred to generate *CDX2P-NLS-Cre;APC*[Δ/+];ROSA-PID mice, termed *CDX2;APC;PID*. Genotyping was performed by PCR analysis of tail biopsy DNA. All mice were examined and weighted once every two weeks for 8 months. Mice were euthanized if mice exhibited signs of illness.

**Cell culture, transfections, and transductions.** Primary colon epithelial cells were established from tumors isolated from *CDX2P-NLS-Cre;APC*[Δ/+];ROSA26-PID and *CDX2P-NLS-Cre;APC*[Δ/+] mice. A human colon cell line, SW48 (ATCC CCL-231, which has been authenticated and tested for mycoplasma free) was used in xenograft experiments. All cell lines were maintained in RPMI-1640 medium supplemented with 10% of FBS, 2 mM L-glutamine and 100 U/ml penicillin/ streptomycin ay 37 °C in a humidified 5% $CO_2$ incubator. Transfection of LNA-miR-200 chimeric LNA/DNA oligonucleotide was carried out with Lipofectamine 2000 (Life Technology) according to the manufacturer's instructions. Cells

expressing PID (#4 and #26) were transfected with either 50 nM of LNA-miR-200 or miRCURY inhibitor control LNA (Exiqon) repeatedly every 3 days for up to a total of 22 days[58]. Non-PID-harboring cells (#35 and #56) were infected with lentivirus constitutively expressing either *miR-200b-200a-429* cluster or *miR-200c-141* cluster[59].

The φNX retrovirus packaging cell line (Orbigen) was transfected using Lipofectamine 2000 (Invitrogen). Viral supernatants were harvested 48 h post-transfection and filtered. Cells were incubated with retroviral supernatant supplemented with 4 μg/ml polybrene for 4 h at 37 °C, and then were cultured in growth media for 48 h for viral integration. Infected cells were selected with 2 μg/ml of puromycin or by flow cytometry for cells with green fluorescent protein (GFP).

Five mouse lentiviral shRNA (TRC0000127334-TRC0000127338; Openbiosystems) were analyzed for knockdown efficiency of ESRP1 in *PID*+ cells using Lipofectamine 2000 (Life Technology) according to the manufacturer's instructions. Total protein lysates were extracted from *PID*+ cells infected by lentivirus with puromycin selection for 2 weeks and evaluated kncoldown efficiency by immunoblots.

**Apoptosis assay**. Cells at $2 \times 10^6$ was trypsized, washed with PBS and fixed overnight in 4% paraformaldehyde. Cells were spinned down and resuspended in HistoGel (Thermo Scientific) and allowed to solidify at room temperature. HistoGel pellet was embedded in parafilm and section was stained with cleaved caspase 3 to analyze apoptosis and hematoxylin and eosin (H&E) stained section was used for diagnostic purposes.

**Cell proliferation assay**. Cells were plated at $2 \times 10^3$ in 96-well plates and 10 μl of 3-[4,5-dimethylthiazol-2-yl]-2,5-diphenyl tetrazolium bromide (MTT) solution was added to each well to a final concentration of 0.5 mg/ml. The reaction was stopped after 4 h at 37 °C by adding 100 μl of solubilization solution (10% SDS in 0.01 M HCl) and the samples were analyzed at 595 nm on Perkin Elmer Envision plate reader. Triplicates were performed for each sample, and experiments were preformed on three occasions.

**Cell motility and invasion assays**. Matrigel invasion and BioCoat control chamber (BD Biosciences) were rehydrated in serum-free RPMI-1640 medium for 2 h and then placed in 0.75 ml of RPMI-1640 medium supplemented with 5% FBS. Cells at a density of $2 \times 10^4$ suspended in 0.5 ml of RPMI-1640, and seeded onto Matrigel chambers. Cells were allowed to migrate for 19 h. Cells on the upper surface were gently removed with a cotton bud, and cells that had migrated through the 8-μm pores were fixed with 4% paraformaldehyde for 15 min and stained with 0.1% crystal violet for 15 min. Membranes were washed, removed and mounted on a glass slide, and the level of invasion was quantified by visual counting using a microscope with a 20x objective. Cell motility was determined by counting cells migrated through the BioCoat control chamber, and cell invasiveness was determined by counting cells invaded through the Matrigel on the 8 μm membrane pores of the Matrigel invasion chamber.

**RNA isolation, (q)RT-PCR and RNA-seq**. Total RNA was isolated using Tri reagent (Sigma) and digested by DNase I (Life Technology). First-Strand cDNA was synthesized using SuperScript III reverse transcriptase (Life Technology) according to manufacture's protocol and standard PCR was performed afterwards. Quantitative Real-time PCR analysis was accomplished using Fast SYBR Green master mix on a 7500 quantitative Real-time PCR machine (Applied Biosystems). The relative amount of gene transcripts and pre-micro-RNA (pre-miRNA) was normalized to GAPDH and U6, respectively. The primers sequences were listed in Supplementary Table 1.

The mouse reference genome assembly (GRCm38.p3) was obtained from Gencode and indexed using Bowtie2 (v2.1.0). The paired end reads were cleaned of adapters using CutAdapt (v1.8.1), and FastQC (v0.11.2) was used to examine read quality. The cleaned reads were aligned using TopHat version 2.0.8 with a gene annotation file of all protein coding genes provided and detection of novel junctions deactivated. For each condition (treated and untreated) three sets of reads from replicated sequencing runs were provided to TopHat, and a single alignment file generated for each condition. These alignment files (BAM) were converted to a text-readable (SAM) format using Samtools version (v0.1.19). One SAM file for each condition was provided to CuffDiff, with compatible hits normalization activated and the gene annotation (GTF) file again provided as input. CuffDiff generated a number of files used to analyze the differential expression between these two conditions. Expression analysis was performed using the CummeRbund (v2.7.2) R package, which takes the differential expression output files from CuffDiff as input.

Pathway analysis was performed on a number of cancer related pathways, which were obtained from the Kegg database. For each pathway we used the cummeRbund package to generate a heatmap of all genes as well as a heatmap for only those genes with significant fold change between conditions. Expression plots were generated for a specific set of genes in order to compare the RNA-seq expression analysis with RT-PCR results for confirmation. The results between

these two methods were highly concordant showing that the expression of these genes was altered by treatment.

**Immunofluorescence**. NCells were plated on glass coverslips in 6-well culture plates and fixed in 4% paraformaldehyde for 10 min, permeabilized with 0.2% Triton X-100 for 10 min and blocked with 1% BSA in PBS for 20 min. The coverslips were incubated with the Snai1 antibody (abcam #180714, 1/100) for 1 h at room temperature. After three washes, the coverslips were incubated with goat anti-rabbit IgG-Alexa-Green 488 secondary antibody (Molecular Probe #A11008, 1/500) for 1 h at room temperature. Nuclei were stained with DAPI (Molecular Probe, 1/5000). Images were observed and captured on an inverted phase/fluorescence microscope with a 40x objective (Nikon TE300).

**Tissue preparation, histology, immunohistochemistry**. All tumor lesions, control tissues and internal organs were fixed overnight in 4% paraformaldehyde, dehydrated and embedded in paraffin. Hematoxylin and eosin stained sections were used for diagnostic purposes and unstained sections for immunohistochemical (IHC) studies. IHC was performed with the following antibodies: rabbit polyclonal antibody for phospho-Pak, phospho-Mek (Life Technology #44940 G, 1/300, and #44460 G, 1/50, respectively),, phospho-Akt, phospho-Histone H3, and cleaved caspase 3 (Cell Signaling Technology #4060, 1/50, #9701, 1/200, and #9661, 1/200 respectively), Msi1 (Santa Cruz Biotechnology #sc-135721, 1/100), Bmi1 (abcam #ab14389, 1/50), ESRP1 (Novus Biologicals #NBP1-82201, 1/200), E-Cadherin and BrdU (BD Transduction Laboratories #610182, 1/100 and #347580, 1/100, respectively). The evaluation of the IHC was conducted blindly, without knowledge of the genotype.

For cytoplasmic P-Pak, P-Mek, P-Akt, Bmi1 and Msi1, positive staining results were scored by a pathologist and interpreted in categories from 0 to 3+ as follows: 0, no staining; 1+, 10–40% of the cells stained; 2+, 40–60%; and 3+, >60%. The number of positive cells for cleaved caspase-3 was counted under high-power microscopic field. For ESRP1 and E-Cadherin, slides were scanned using an Aperio ScanScope CS 5 slide scanner. The positive nuclear percentage for ESRP1 was quantified using the the Aperio nuclear V9 algorithm and the intensity score for E-Cadherin was quantified using the Aperio membrane V9 algorithm.

**Immunoblotting**. Western blot analyses were performed on lysates extracted from primary cell cultures with RIPA lysis buffer (50 mM Tris-HCl, 150 mM NaCl, 1% NP40, 0.25% sodium deoxycholate, 1 mM PMSF, protease inhibitor and phosphase inhibitor cocktail). Protein concentrations were determined, and equal amounts of total proteins were separated on SDS-PAGE. Antibodies used included anti-Zeb1 (#3396; 1/500), -Slug (#9585; 1/500), -Twist (#46702; 1/500), -Fibronectin (#6238; 1/500), -Pak1 (#2602; 1/2000), -Pak2 (#2608; 1/1000), -Pak3 (#2609; 1/1000), -Mek (#9122; 1/2000), -Erk (#9101; 1/5000), -phospho-Erk1/2 (Thr202/Tyr204) (#9101; 1/5000), -Akt (#9272; 1/1000), -phospho-Akt (Ser473) (#4060; 1/1000), -mTOR (#2983; 1/1000), -phospho-mTOR (Ser2448) (#2971; 1/1000), -phospho-β-Catenin (Ser675) (#4176; 1/1000), -GST (#2622; 1/1000), and -cyclin D1 (#2978; 1/1000), all from Cell Signaling Technology; anti-phospho-Pak (Ser141) (#44940G; 1/7000) and -phospho-Mek (Ser298) (#44460G; 1/2000) were from Life Technology. Vinculin (#V9131; 1/1000) was from Sigma-Aldrich. Antibodies against β-Catenin (#610154; 1/2000), E-Cadherin (#610182; 1/1000) and N-Cadherin (#610920; 1/1000) were from BD Transduction Laboratories; Occludin (#sc-5562; 1/1000), PCBP1 (#sc-393076; 1/1000), CD44 (#sc-18849; 1/300), and Lamin A/C (#sc-7292; 1/1000) were from Santa Cruz Biotechnology; hnRNPM (#TA301557; 1/1000); ESRP1 (#NBP1-82201; 1/1000) was from Novus Biologicals; Snai1 (#ab180714; 1/500) and phospho-Snai1 (#ab63568; 1/250) were from Abcam. GAPDH (#2118; 1/10000) or Actin (#3700; 1/10000) was used as loading controls. Uncropped images of key western blots are available in Supplementary Fig. 9.

**Tumorigenicity assays**. Cells were trypsinized, washed with PBS and resuspended at $10^7$ cells per ml in PBS. $2 \times 10^6$ cells were injected subcutaneously into flanks of 6-week old nude mice (BALB/c nu/nu). Mice were monitored and tumor diameters were measured every 2 days using caliper for 2 weeks following injection. Tumor volume was calculated by the following formula: volume = 0.5 × (length) x (width)$^2$. The mice were sacrificed and the tumors were resected for histological examination and immunoblot analysis.

**Treatment with Pak inhibitors**. Frax1036 ($C_{28}H_{32}ClN_7O$; M.W. 518.05) was formulated in 20% of 2-Hydroxypropyl-β-cyclodextrin in 50 mM citrate buffer (pH 3.0) and administrated to mice receiving either single dose of 30 mg/kg/day or an equivalent volume of vehicle via oral gavage. G-5555 ($C_{25}H_{25}ClN_6O_3$; M.W. 492.96) was reconstituted in corn oil and MCT (0.5% (w/v) methylcellulose/0.2% (w/v) Tween 80 in sterile water) and administrated to mice twice either dose of 25 mg/kg/day or vehicle by oral gavage, with the second dose given 6 h interval. All treatments were continued for 2 weeks, at which time the animals were sacrificed.

**Image quantification**. All immunoblots, immunofluorescence images, and RT-PCR gel images were obtained by at least three independent experiments and quantified by a freeware Image J v1.5 downloaded from NIH website. Statistical

differences in expression between *PID*+ and *PID*- cells were presented as mean ± SEM using Student's *t*-test.

**Statistical analysis**. Statistical analysis was performed using GraphPrism. Tumor numbers and average size were compared with control groups using a two-tailed Student's *t*-test with *P* < 0.05 considered statistically significant. *P < 0.05, **P < 0.005, ***P < 0.0005, student *t*-test. NS, not significant.

**Data availability**. All materials are available to the research community. A material transfer agreement is required to obtain the ROSA26-LSL-PID mice. Data that support the findings of this study have been deposited in the NCBI Gene Expression Omnibus (GEO; https://www.ncbi.nlm.nih.gov/geo/) with the accession number GSE116832 and all relevant data are available from the authors upon reasonable request.

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

## Acknowledgements

We thank Drs. Jody Haigh for providing *Rosa26*-targeting plasmids and advice and Eric Fearon for advice on the mouse colon cancer model, and the Fox Chase Cancer Center Animal Facility for assistance with mouse experiments. This work was supported by CA142928 and CA148805 (to J.C.), CORE Grant P30 CA006927 and an appropriation from the state of Pennsylvania to the Fox Chase Cancer Center.

## Author contributions

H.Y.C. constructed the targeting vectors, characterized the ROSA26-LSL-PID mice, performed experiments listed in all the figures, and wrote the draft for the manuscript. B.D. performed mRNA array experiments and did statistical analysis. J.N.K. perfomed immunoblots and assisted with data interpretation. C.T.Z. performed quantification of immunofluorescence. T.Y.P. constructed ESRP1 vector. C.A.V. analyzed mRNA expression. J.C. assisted with experimental design and interpretation, and preparation of the manuscript.
