## [Peer Review File · Nature Communications]

Reviewers' comments:

Reviewer #1 (Remarks to the Author):

NCOMMS-17-23062

General comments

This is an interesting study using of an inhibitory peptide to inhibit the kinase activity of all three group I Pak family members (Pak1-3), leading to prevention of colorectal cancer in vivo. The authors put a strong emphasis on the potential of targeting Pak1 as a treatment option for colon adenocarcinoma. However, while the utilized model indicates that targeting of Pak could prevent tumor formation, it does not actually address Pak targeting as a treatment option - this would instead require Pak inhibition at the stage of already developed adenocarcinoma. In addition, authors need to control for the Pak specificity of the used PID peptide and perform a number of additional controls, including proper quantifications. Further, the analysis of the tumor tissue is very sparse and further such analysis will be essential.

If these issues could be experimentally addressed, this study could significant contribute to our understanding of Pak kinases in cancer and support the notion of Pak targeting as a colon cancer treatment option.

Specific points

1. The authors emphasize the potential of targeting Pak as a treatment option for colon cancer, although this study so far is limited to a cancer preventive effect of PID overexpression. To prove their point, authors should treat their mice with fully developed adenocarcinoma in vivo to test the outcome. Against this, the authors use the claim that available Pak inhibitors display toxicity as a major obstacle for such an approach. However, several previous studies have utilized Pak inhibitors for in vivo treatment, including in a recent paper where the lead author of this MS is a co-author; where existing Pak inhibitors were used for in vivo treatment of melanoma (Lu et al. Nature 2017). Consequently, Pak inhibitors could be used at least for short term treatments of mice in vivo and could be utilized here to test their impact on colon adenocarcinoma.

2. Specificity concerns: The authors use an unusually large tag (GST; around 30 kD) to express the PID peptide. Use of large tags raises concerns about potential off-target effects by the tag itself. GST also comprise an intrinsic enzyme activity, raising further such concerns. The authors need to control for potential effects of the GST tag alone. Further, the lead author previously published a study indicating off-target effects of the PID peptide to inhibit cell proliferation (Ref #39). These off-target effects are here suggested to depend on PID interaction with FXR1/FMR1 and the here used PID sequence was therefore mutated to prevent FXR1/FMR1 binding. However, it is not clear if the FXR1/FMR1 actually accounts for the previously observed off-target effects; this also needs to be controlled for.

To control for potential GST and PID off-target effects, authors could transfect their PID-negative cells to compare the effect on cell proliferation of: Mock; GST alone; GST-PID-E129K (used here in mice); and GST-PID-E129K/L107F (preventing both FXR1/FMR1 and Pak interactions).

3. The expression of the Pak inhibitory peptide (PID) is present already at the start of tumor development. A very interesting observation is that stem cell markers were downregulated in isolated cells (SF4B). This suggests that the observed colon cancer prevention may be due to depletion of stem cells from the colon (and/or inhibition of stem cell properties), thereby depriving the tissue from tumor originating cells. However, this finding appears to be underemphasized and insufficiently analyzed in vivo. To better clarify this, the authors should stain the colon tissues (normal, adenoma and adenocarcinoma tissues from wt and PID expressing mice) for these stem cell markers.

4. It also appears surprising that such potential stem cell deprivation would not have any impact on colon development as suggested by SF1A. Firstly, the tissue stainings in SF1 needs to be properly quantified based on several mice per group. Secondly, if there would be no effect on colon development, it should be discussed how the colon may develop normally if stem cells are deprived or compromised.

5. The mechanistic efforts rely heavily on cell culture, while tissues remain under analyzed.

Available adenoma tissues should be analyzed for in situ: 1) Epithelial and mesenchymal markers to substantiate the conclusions on EMT; 2) GST, to verify GST-PID expression in the tumors; 3) Ki67 and Caspase-3 for the effects on cell growth in vivo 4) miR200; ESPR1; phospho-Snai; and other key signaling events detected in vitro to verify that these signaling events are affected by PID in tumors.

6. Authors claim that an EMT program is mediated by miR200; alternative CD44 splicing; the splicing factor ESPR1; and phosphorylation of Snai. However, the mediating function is only examined for miR200, while claims about other mediating factors need to be supported by experimental evidence testing directly their function as mediators of EMT in the used models.

7. The legend for F1C is missing.

8. F1F lacks blotting for total Pak1 and also lacks quantification.

9. F3A claims to display cell viability, although the actual analysis is of the total number of mitochondria in the cell population (MTT). To measure viability, authors will need to analyze apoptosis and other potential causes of cell death.

10. All immunoblots (and qPCR gels) need to be properly quantified (based on exposures within the quantitative linear range) and each be based on at least three independent experiments.

11. All the tissue stainings need to be quantified, both for the morphology and in particular the markers (signaling, etc.) labelled by antibodies.

12. All the cell stainings (e.g. SF5E) must be also be properly quantified.

12. FS4. A more informative heat map should be provided and identify the DE genes. Also, an Ontology enrichment analysis of DE genes should be presented.

Minor point

1. Typos in need of correction: p. 1: target; FS4: Kidney.

Reviewer #2 (Remarks to the Author):

In this manuscript, the authors investigate the effect of inhibiting Pak-1 in a mouse model using an inducible form of a peptide that constitutes the auto-inhibitory domain of Pak-1. They show that expressing this peptide specifically in the distal intestinal and colonic epithelium decreases the severity of the tumour phenotype of Apc mutant mice.

Although the manuscript is overall well written and the data presented clearly, I have a number of reservations/queries.

If Pak-1 is required for activation of Wnt signalling (as stated in the introduction) then its inhibition should be deleterious for normal tissue homeostasis in the intestinal crypt. However, that is not what is observed, challenging the idea that Pak-1 is indeed required for Wnt signalling (which is crucial for normal crypt homeostasis). This requires an explanation and also determining if the expression of the PID causes any changes in the normal epithelium in greater detail. For instance, is PID expressed in ALL cells when induced?

The statement that APC-null cells assume stem cell characteristics (line 217/218) is not accurate. They behave more like transit amplifying cells in that they continually divide. Normal stem cells cycle slowly, not rapidly.

Has it been established how scaffolding functions of Pak-1 are affected by the PID?

The nature of the mutant Apc mouse is not explained and only referenced late in the text. This is not helped by the nomenclature changing from APC Δ /+ to APC null, to APCloxP/+. That means the nature of this particular model is difficult to know. I assume it is the same mouse as described in Hinoi et al 2007, but this needs to be made much clearer early on. Particularly, since in most APC mutant mouse models, tumours arise predominantly in the small intestine. In the model used

here, tumours seem to arise mostly in the colon. This needs to be explained much better to allow comparison to existing studies.

Invasiveness and motility are mentioned in the results (Line 179/180) but in the methods only 'motility' assays are described.

The conclusion that Pak-1 directly regulates Snai1 is not supported by the data. Lack of active Pak1 correlated with decreased Snai1 phosphorylation, that does not mean Pak-1 is the direct kinase.

What is the explanation for the cohort that completely lacks tumours?

Which animals were used for the analysis shown in Figures 3-5? Those without any tumours? That should be stated explicitly. Would tissue from APC/PID mice that did develop tumours yield different results?

In summary, although the findings are interesting and indicate a key function of Pak-1 in tumour progression in an APC mutant model, there are many details that need to be explained and/or considered to allow better understanding of the results and their implication.

Reviewer #1

1. *The authors emphasize the potential of targeting Pak as a treatment option for colon cancer, although this study so far is limited to a cancer preventive effect of PID overexpression. To prove their point, authors should treat their mice with fully developed adenocarcinoma in vivo to test the outcome. Against this, the authors use the claim that available Pak inhibitors display toxicity as a major obstacle for such an approach. However, several previous studies have utilized Pak inhibitors for in vivo treatment, including in a recent paper where the lead author of this MS is a co-author; where existing Pak inhibitors were used for in vivo treatment of melanoma (Lu et al. Nature 2017). Consequently, Pak inhibitors could be used at least for short term treatments of mice in vivo and could be utilized here to test their impact on colon adenocarcinoma.*

We appreciate these comments, but had to take a different approach than that suggested by the reviewer to address this important concern. The mouse model used in our manuscript is not well-suited for preclinical drug studies, because the *CDX2P-Cre;Apc^{lox/+}* mice take about 10 months to develop CRC (and they don't all do so synchronously), at which point, one would need to start treating relatively large cohorts for at least several weeks. Even then, I'm not sure there would be enough statistical power to prove a point. Investigators have indeed used CRC GEMM models for preclinical studies, but such experiments are most conveniently done in a more synchronized model such as *CDX2P-CreER^{T2};Apc^{fl/fl}* mice, in which both alleles of *Apc* are deleted in the colon upon tamoxifen administration. Such mice rapidly develop CRC and have been used to test the antiproliferative effects of drugs such as rapamycin (Hardiman *et al.*, PLoS One, e96023, 2014). The recently published melanoma studies cited by the reviewer did not face such steep logistical hurdles as we would face using our *CDX2P-Cre;Apc^{lox/+}* mice.

Instead, in the revised manuscript we used a xenograft approach, which has the virtue of displaying relatively synchronous, observable tumor growth. We realize that the xenograft approach has its own limitations, but believe that they get at the issue of treating established disease with Pak inhibitors, as might be encountered in CRC patients. We hope the reviewer accepts the need for and validity of this approach.

2. *Specificity concerns: The authors use an unusually large tag (GST; around 30 kD) to express the PID peptide. Use of large tags raises concerns about potential off-target effects by the tag itself. GST also comprises an intrinsic enzyme activity, raising further such concerns. The authors need to control for potential effects of the GST tag alone. Further, the lead author previously published a study indicating off-target effects of the PID peptide to inhibit cell proliferation (Ref #39). These off-target effects are here suggested to depend on PID interaction with FXR1/FMR1 and the here used PID sequence was therefore mutated to prevent FXR1/FMR1 binding. However, it is not clear if the FXR1/FMR1 actually accounts for the previously observed off-target effects; this also needs to be controlled for. To control for potential GST and PID off-target effects, authors could transfect their PID-negative cells to compare the effect on cell proliferation*

of: Mock; GST alone; GST-PID-E129K (used here in mice); and GST-PID-E129K/L107F (preventing both FXR1/FMR1 and Pak interactions).

We agree with the reviewer and now include the suggested experiments to rule out significant contributions of the GST moiety and also non-Pak related interactions. These data are presented in the Results section and shown in Supplementary Fig. S5A and S5B.

3. *The expression of the Pak inhibitory peptide (PID) is present already at the start of tumor development. A very interesting observation is that stem cell markers were downregulated in isolated cells (SF4B). This suggests that the observed colon cancer prevention may be due to depletion of stem cells from the colon (and/or inhibition of stem cell properties), thereby depriving the tissue from tumor originating cells. However, this finding appears to be underemphasized and insufficiently analyzed in vivo. To better clarify this, the authors should stain the colon tissues (normal, adenoma and adenocarcinoma tissues from wt and PID expressing mice) for these stem cell markers.*

We now present data regarding stem cell markers in the colon tissues. These IHC data are presented in Supplementary Fig. S7C. They show a reduction in Msi1 (and possibly Bmi1) expression in CRC tissue from mice expressing the PID.

4. *It also appears surprising that such potential stem cell deprivation would not have any impact on colon development as suggested by SF1A. Firstly, the tissue stainings in SF1 needs to be properly quantified based on several mice per group. Secondly, if there would be no effect on colon development, it should be discussed how the colon may develop normally if stem cells are deprived or compromised.*

We now present quantifications for Supplementary Fig. S1 as well as discussion around the issue of colon development in the PID mice. SF1A shows that PID expression does not reduce the number of BrDU positive cells. We assume that the reduction in Wnt-signaling by the PID is not severe enough to affect normal development, but is enough to delay/reduce tumor formation. There are some examples in the literature that support this idea. The most relevant paper is from Fearon's group, showing that hemizygous loss of beta-catenin suppressed tumorigenesis in a *Apc* model of CRC, but did not affect development or function of the intestines in *Apc* WT mice (Feng *et al.* PLOS Genet. 11, 2015). Our PID-expressing mice may likewise reduce, but not eliminate, beta-catenin signaling. Other supporting examples from the literature include a paper from Naumov *et al.* who showed that loss of *CD24* reduced Wnt signaling and suppressed intestinal tumorigenesis in *Apc* mutant mice but did not affect tissue homeostasis (*Int. J. Cancer*, 135:1048, 2014). Similar results were seen in *Id1* knock mice (*Cancer Prev. Res.* 8:303, 2015). Also, long-term treatment with Wnt pathway inhibitors such as axitinib was reported to block tumor formation in cancer cells, zebrafish, and *Apc* min/+ mice, but did not affect intestinal homeostasis (Qu *et al.*, PNAS 113:9339, 2016). These points are made explicit in the revised discussion.

5. *The mechanistic efforts rely heavily on cell culture, while tissues remain under analyzed. Available adenoma tissues should be analyzed for in situ: 1) Epithelial*

and mesenchymal markers to substantiate the conclusions on EMT; 2) GST, to verify GST-PID expression in the tumors; 3) Ki67 and Caspase-3 for the effects on cell growth in vivo 4) miR200; ESPR1; phospho-Snai; and other key signaling events detected in vitro to verify that these signaling events are affected by PID in tumors.

We now present extensive IHC data on these tumors. In some cases, IHC-grade antibodies were not currently available (e.g., anti p-Snai) or did not stain properly (anti-GST), but we were able to obtain data for markers in each category, including Msi, Bmi1, ESRP1, E-Cadherin, and Caspase 3). These data are presented in Supplementary Fig. S7.

6. *Authors claim that an EMT program is mediated by miR200; alternative CD44 splicing; the splicing factor ESPR1; and phosphorylation of Snai. However, the mediating function is only examined for miR200, while claims about other mediating factors need to be supported by experimental evidence testing directly their function as mediators of EMT in the used models.*

We now show that manipulating ESRP levels with shRNA leads to corresponding changes in CD44 splicing (Fig. 5E). We also tried a similar functional analysis of CD44 by expressing CD44v in PID- cells (using a vector from Toru Hiraga), but this vector failed to alter CD44v levels. We could not source any other CD44v expression vectors.

7. *The legend for F1C is missing.*

This issue has been fixed.

8. *F1F lacks blotting for total Pak1 and also lacks quantification.*

This issue has been fixed.

9. *F3A claims to display cell viability, although the actual analysis is of the total number of mitochondria in the cell population (MTT). To measure viability, authors will need to analyze apoptosis and other potential causes of cell death.*

We now present data regarding apoptosis in these colon cell lines and are presented in Supplementary Fig. S3B.

10. *All immunoblots (and qPCR gels) need to be properly quantified (based on exposures within the quantitative linear range) and each be based on at least three independent experiments.*

This issue has been addressed.

11. *All the tissue stainings need to be quantified, both for the morphology and in particular the markers (signaling, etc.) labelled by antibodies.*

This issue has been addressed.

11. *All the cell stainings (e.g. SF5E) must be also be properly quantified.*

This issue has been addressed.

12. *FS4. A more informative heat map should be provided and identify the DE genes. Also, an Ontology enrichment analysis of DE genes should be presented.*

The heat map has been updated and the DE genes identified. In addition, we now present GO enrichment data.

Minor point

1. *Typos in need of correction: p. 1: target; FS4: Kidney.*

This issue has been addressed.

Reviewer #2

If Pak-1 is required for activation of Wnt signalling (as stated in the introduction) then its inhibition should be deleterious for normal tissue homeostasis in the intestinal crypt. However, that is not what is observed, challenging the idea that Pak-1 is indeed required for Wnt signalling (which is crucial for normal crypt homeostasis). This requires an explanation and also determining if the expression of the PID causes any changes in the normal epithelium in greater detail. For instance, is PID expressed in ALL cells when induced?

See point #4 above. Unfortunately, we were not able to use IHC to document PID expression on a cell-by-cell basis. We believe that PID expression leaves enough residual Wnt signaling to support normal tissue homeostasis, but not enough for effective tumorigenesis, similar to the argument made by Fearon's group in Feng *et al.* (PLOS Genet. 11, 2015).

The statement that APC-null cells assume stem cell characteristics (line 217/218) is not accurate. They behave more like transit amplifying cells in that they continually divide. Normal stem cells cycle slowly, not rapidly.

We have edited the text to address this point.

Has it been established how scaffolding functions of Pak-1 are affected by the PID?

That is an interesting point. In truth, the answer is "not really." It is assumed that binding to SH3-domain containing proteins like PIX or other binders such as PP2A and/or POPX are unaffected, but I don't think anyone has really looked into this issue. We added some discussion around this point in the revised manuscript.

*The nature of the mutant Apc mouse is not explained and only referenced late in the text. This is not helped by the nomenclature changing from APC^{Δ/+} to APC null, to APCloxP/+. That means the nature of this particular model is difficult to know. I assume it is the same mouse as described in Hinoi *et al* 2007, but this needs to be made much clearer early on. Particularly, since in most APC mutant mouse models, tumours arise predominantly in the small intestine. In the model used here, tumours seem to arise mostly in the colon. This needs to be explained much better to allow comparison to existing studies.*

The model is indeed the one described in Hinoi *et al.* 2007. In the revised manuscript, we now name the model explicitly early on and discuss its properties.

Invasiveness and motility are mentioned in the results (Line 179/180) but in the methods only 'motility' assays are described.

We have added the methodology for invasion along with motility in Materials and Methods.

The conclusion that Pak-1 directly regulates Snai1 is not supported by the data. Lack of active Pak1 correlated with decreased Snai1 phosphorylation, that does not mean Pak-1 is the direct kinase.

We agree with the reviewer. There is a correlation, which is not the same thing as proving causation. However, Pak has been implicated as a direct kinase for Snai1 at this site, so the results suggest that a straightforward relationship might exist. We have softened the text regarding these results.

What is the explanation for the cohort that completely lacks tumours?

Within the limits of the experiment depicted in Fig. 2, it appears that expression of the PID allows for the occasional development of adenomas (2C), but does not allow for progression to carcinoma (2D). We assume that the loss of key signaling pathways (Erk, Akt, b-catenin) underlies this phenomenon).

Which animals were used for the analysis shown in Figures 3-5? Those without any tumours? That should be stated explicitly. Would tissue from APC/PID mice that did develop tumours yield different results?

These cells came from animals with adenomas, as in Fig. 2C (used because none of the PID expressing animals developed adenocarcinomas). We now state this explicitly.

Sincerely,

Jonathan Chernoff

REVIEWERS' COMMENTS:

Reviewer #1 (Remarks to the Author):

The authors have improved this manuscript significantly and have addressed most of my concerns. However, there are a few issues that need to be addressed before I would recommend publication.

1. CRC xenograft experiments have been added with treatment using two Pak inhibitors. The authors conclude: "dramatic tumor regression was observed in mice treated with Frax-1036". However, this is not true. The starting volume is specified to 40 mm³ and the average end volume to 100 mm³. Thus, there is no tumor regression, but only a suppression of tumor growth.
2. While several quantifications have been added to address my initial concerns, quantification of the stainings in SFig 7C is lacking.
3. The in vitro experiments indicate that PID blocks cell proliferation. However, in the APC model, there appears to be no difference in proliferation or apoptosis (FigS1), but still a delay in tumor development. Then, in the CRC xenograft model, there appears to be a reduction of tumor growth by treatment with Pak inhibitors (FigS4), but it was not assessed if this may be due to changed proliferation and/or apoptosis. The different outcomes between the different models need to be discussed.

Reviewer #2 (Remarks to the Author):

Overall the revisions were very thorough and addressed my queries. I particularly appreciated the extremely clear language of the response and only have a few minor suggestions:

Line 95-98: I feel this was slightly overstated and suggest changing the language to:
"By analyzing colonic epithelial cells derived from these mice, we demonstrated the existence of a Group I Pak-regulated EMT program in Apc-mutant CRC cells, **that involves** the miR-200 microRNA family, expression of the CD44 splicing factor ESRP1, and phosphorylation of the Snai transcription factor."

Please correct line 390: B-catenin is not a transcription factor itself. It regulates the activity of transcription factors, it is a transcriptional regulator.

I still could not find the migration assay description, only the Matrigel Invasion assay.

The preparation of the cell lysates was also not described, or at least I could not find it in the relevant section in the methods.

One thing that is not really discussed, other than a mention in the introduction, is how the effects of PAK-1 on cytoskeletal dynamics may contribute to the observed effects. At least a small paragraph about this should be included in the discussion.

Lastly, and I realise that it is not fair to introduce this point at this stage of the review process and thus I could understand if it was ignored, but I wanted to raise it anyway so see if it could be considered. All the comparisons of the gene profiles shown in the Supplemental Figures, are between PID+ and PID- situations. Clearly, this is the point of the study and is being used to support the relevant observation. However, it would be very helpful to know how the expression of the genes shown in the supplemental Figures for both situations relate to those in completely wild type samples. It could help to know which changes may be more relevant than others. These

comparisons may already exist in the literature making it relatively straightforward to comment on what is specific to this model and what may be related to the PID effects?

Reviewer #1

1. *CRC xenograft experiments have been added with treatment using two Pak inhibitors. The authors conclude: “dramatic tumor regression was observed in mice treated with Frax-1036”. However, this is not true. The starting volume is specified to 40 mm³ and the average end volume to 100 mm³. Thus, there is no tumor regression, but only a suppression of tumor growth.*

We agree, and have changed the text accordingly.

2. *While several quantifications have been added to address my initial concerns, quantification of the stainings in SFig 7C is lacking.*

We apologize for the omission, which has now been fixed.

3. *The in vitro experiments indicate that PID blocks cell proliferation. However, in the APC model, there appears to be no difference in proliferation or apoptosis (FigS1), but still a delay in tumor development. Then, in the CRC xenograft model, there appears to be a reduction of tumor growth by treatment with Pak inhibitors (FigS4), but it was not assessed if this may be due to changed proliferation and/or apoptosis. The different outcomes between the different models need to be discussed.*

We now discuss this point. It seems the major effect of Pak blockade is on proliferation rather than survival, consistent with loss of ERK and catenin signaling.

Reviewer #2

*Line 95-98: I feel this was slightly overstated and suggest changing the language to: “By analyzing colonic epithelial cells derived from these mice, we demonstrated the existence of a Group I Pak-regulated EMT program in Apc-mutant CRC cells, **that involves the miR-200 microRNA family, expression of the CD44 splicing factor ESRP1, and phosphorylation of the Snai transcription factor.**”?*

We agree and have altered the text accordingly.

Please correct line 390.: B-catenin is not a transcription factor itself. It regulates the activity of transcription factors, it is a transcriptional regulator..

As above, we agree and have altered the text accordingly.

I still could not find the migration assay description, only the Matrigel Invasion

assay.

The preparation of the cell lysates was also not described, or at least I could not find it in the relevant section in the methods.

We have added the methodology for invasion along with motility in Materials and Methods and have now included methods for cell lysis.

One thing that is not really discussed, other than a mention in the introduction, is how the effects of PAK-1 on cytoskeletal dynamics may contribute to the observed effects. At least a small paragraph about this should be included in the discussion.

That is true. Some of the effects of Pak blockade may well be due to loss of cytoskeletal signaling. As we did not examine this possibility in detail in our experiments, we didn't discuss it in detail in the paper. However, we now include a few new sentences in the Discussion to address this important possibility.

Lastly, and I realise that it is not fair to introduce this point at this stage of the review process and thus I could understand if it was ignored, but I wanted to raise it anyway so see if it could be considered. All the comparisons of the gene profiles shown in the Supplemental Figures, are between PID+ and PID- situations. Clearly, this is the point of the study and is being used to support the relevant observation. However, it would be very helpful to know how the expression of the genes shown in the supplemental Figures for both situations relate to those in completely wild type samples. It could help to know which changes may be more relevant than others. These comparisons may already exist in the literature making it relevelatively straightforward to comment on what is specific to this model and what may be related to the PID effects?

The reviewer raises a good point, and one that we had considered while the experiments were ongoing. The issue here is in obtaining a comparable data set. At the Jackson Lab Gene Expression Data (GXD) site (<http://www.informatics.jax.org/expression.shtml>), there are data regarding transcript expression from C57BL/6 mouse colonic epithelium, but these data are derived from cells that were obtained directly from the colon and lysed without culturing. In contrast, our PID+ and PID- cells were primary cultures from the adenomas formed in the *Apc* mouse model. In retrospect, perhaps we ought to have made an additional primary colonic epithelial line from normal mouse colonic epithelia, but those still wouldn't be from adenomas, as none would form in WT mice.

Sincerely,

Jonathan Chernoff